# Kaposi’s Sarcoma-Associated Herpesvirus Reactivation by Targeting of a dCas9-Based Transcription Activator to the ORF50 Promoter

**DOI:** 10.3390/v12090952

**Published:** 2020-08-27

**Authors:** Endrit Elbasani, Francesca Falasco, Silvia Gramolelli, Veijo Nurminen, Thomas Günther, Jere Weltner, Diego Balboa, Adam Grundhoff, Timo Otonkoski, Päivi M. Ojala

**Affiliations:** 1Translational Cancer Medicine Research Program, Faculty of Medicine, University of Helsinki, 00290 Helsinki, Finland; francesca.falasco@gmail.com (F.F.); silvia.gramolelli@helsinki.fi (S.G.); veijo.nurminen@helsinki.fi (V.N.); 2Heinrich Pette Institute, Leibniz Institute for Experimental Virology, 20251 Hamburg, Germany; thomas.guenther@leibniz-hpi.de (T.G.); adam.grundhoff@leibniz-hpi.de (A.G.); 3Stem Cells and Metabolism Research Program, Faculty of Medicine, University of Helsinki, 00290 Helsinki, Finland; jere.weltner@ki.se (J.W.); diego.balboa@crg.eu (D.B.); timo.otonkoski@helsinki.fi (T.O.); 4Department of Infectious Diseases, Imperial College London, London W2 1NY, UK

**Keywords:** KSHV, dCas9, CRISPRa, DD-dCas9-VP192, ORF50, RTA, KSHV lytic cycle, KSHV reactivation, Kaposi’s sarcoma-associated herpesvirus

## Abstract

CRISPR activation (CRISPRa) has revealed great potential as a tool to modulate the expression of targeted cellular genes. Here, we successfully applied the CRISPRa system to trigger the Kaposi’s sarcoma-associated herpesvirus (KSHV) reactivation in latently infected cells by selectively activating ORF50 gene directly from the virus genome. We found that a nuclease-deficient Cas9 (dCas9) fused to a destabilization domain (DD) and 12 copies of the VP16 activation domain (VP192) triggered a more efficient KSHV lytic cycle and virus production when guided to two different sites on the ORF50 promoter, instead of only a single site. To our surprise, the virus reactivation induced by binding of the stable DD-dCas9-VP192 on the ORF50 promoter was even more efficient than reactivation induced by ectopic expression of ORF50. This suggests that recruitment of additional transcriptional activators to the ORF50 promoter, in addition to ORF50 itself, are needed for the efficient virus production. Further, we show that CRISPRa can be applied to selectively express the early lytic gene, ORF57, without disturbing the viral latency. Therefore, CRISPRa-based systems can be utilized to facilitate virus–host interaction studies by controlling the expression of not only cellular but also of specific KSHV genes.

## 1. Introduction

Kaposi’s sarcoma-associated herpesvirus (KSHV) has been etiologically linked to three malignancies, namely Kaposi sarcoma (KS), primary effusion lymphoma (PEL), the plasmablastic variant of multicentric Castleman disease (MCD) and also to two systemic inflammatory diseases: KSHV-inflammatory cytokine syndrome (KICS) and KS immune reconstitution syndrome (KS-IRIS) [1,2,3,4,5]. While PEL is a B cell tumor in which the virus is found predominantly in a latent form, KSHV lytic reactivation is involved in the pathogenesis of the other KSHV-related diseases [6,7,8,9]. Therefore, profound understanding of the KSHV reactivation is required for the successful management of the KSHV-related diseases.

KSHV lytic cycle is initiated by the expression of the replication and transcription activator (RTA) protein encoded by the *ORF50* viral gene [9,10]. It has been demonstrated that *ORF50* expression is necessary and sufficient to trigger the complete lytic replication cycle which culminates with the production and release of infectious viral progeny [9,10]. In culture, KSHV establishes latency in virtually all cell types with a few exceptions. As for instance, in lymphatic endothelial cells KSHV undergoes asynchronous and spontaneous lytic reactivation in a subset of infected cells resulting in the production of high titers of infectious virus [11,12].

In the past years, different chemicals such as TPA (12-O-Tetradecanoylphorbol 13-acetate) and sodium butyrate have been used to break the latency and trigger the KSHV lytic cascade [13,14]. However, these substances have broad effects on the cellular physiology, TPA being an activator of protein kinase C and of NF-kB, and sodium butyrate being a broad histone deacetylase inhibitor. A major advance for studies on KSHV lytic replication has been the generation of two systems: the PEL-derived BCBL1-TREX-RTA and the cancer-derived iSLK.219 [15,16]. Both these cell lines rely on the ectopic expression of ORF50 from a doxycycline-inducible promoter to trigger the KSHV lytic replication cycle. Regardless, to date, we are still missing cellular systems where the KSHV lytic cycle can be activated by governing the *ORF50* expression directly from the viral genome.

CRISPR activation (CRISPRa) systems are derived from the Streptococcus pyogenes CRISPR-Cas9, which can introduce double strand breaks when targeted to specific DNA sites by guide RNAs [17,18]. In the CRISPRa, the Cas9 enzyme is devoid of nuclease activity due to point mutations in the nuclease domain (dead Cas9 = dCas9) and it is fused to transcriptional activators. The use of this engineered, repurposed dCas9, guided to bind at promoters by single-guide RNAs (sgRNAs) has allowed modulation of targeted gene expression [18,19,20]. Furthermore, this highly versatile system has also been used to simultaneously switch on the expression of several genes to drive cellular reprogramming [21,22,23].

In this study, we have used a dCas9 fused to multiple copies of the herpes simplex 1 (HSV-1) VP16 activator domain to trigger the activation of the lytic *ORF50* and *ORF57* gene expression in latently KSHV-infected cells. We show that the dCas9-activator protein can efficiently activate both viral genes. While *ORF50* gene expression activation triggered the full KSHV lytic cycle and virus production, *ORF57* activation did not disturb viral latency.

## 2. Materials and Methods

### 2.1. Cells and DNA Transfection

HEK293 (a kind gift from Tomi Mäkelä, University of Helsinki, Finland) and U2OS (ATCC, Manassas, VA, USA) cells were maintained in DMEM supplemented with 10% fetal calf serum, L-glutamine and Penicillin/Streptomycin. To transfect plasmid DNA into the cells, Fugene 6 HD (Promega, Madison, WI, USA) or JETOptimus (Polyplus, Illkirch, France) were used following instructions provided by the manufacturer. Where indicated, trimethoprim (TMP, Sigma, St. Louis, MO, USA) was added to stabilize DD-dCas9-VP192 protein.

### 2.2. Generation of DNA Constructs

*ORF50* was amplified from KSHV-BAC16 [24] with primers RTAins-Fw and -Rv (5′-aggacctgggctcaggatctgcgcaagatgacaagggtaagaagcttcggcggtc-3′; 5′-gatatcgaattcggcgcgcctcagtctcggaagtaattacgcc-3′) and the pFUW-myc backbone was amplified from pFUW-Myc-mCherry [25] by PCR and using primers pFUW-Myc-BB-Fw and -Rv (5′-agatcctgagcccaggtcctc-3′; 5′-ggcgcgccgaattcgatatc-3′). The DNA fragments were fused together using the HiFi assembly master mix (NEB, Ipswich, MA, USA) and transformed in competent bacteria.

sgRNA spacer sequences that bind to ORF50 and ORF57 promoters were designed using the tools in CRISPR website (www.crispr.mit.edu). The generation of the DNA fragment containing U6 promoters and sgRNA was conducted as described in Balboa et al. [21] using the following sequences as sgRNA oligos (underlined–sgRNA spacer):sg50-1(5′-gtggaaaggacgaaacaccgtcatctccaatacccggaatgttttagagctagaaatag-3′)sg50-2(5′-gtggaaaggacgaaacaccggttcagtcacatgtacgctagttttagagctagaaatag-3′)sg50-3(5′-gtggaaaggacgaaacaccgatgagtcgccggtagctgccgttttagagctagaaatag-3′)sg50-4(5′-gtggaaaggacgaaacaccgccgcccagaaaccagtagctgttttagagctagaaatag-3′)sg57-1(5′-gtggaaaggacgaaacaccgccaaaatagcccgcggcatagttttagagctagaaatag-3′)sg57-2(5′-gtggaaaggacgaaacaccgcacggcccatttttcgtttggttttagagctagaaatag-3′)sg57-3(5′-gtggaaaggacgaaacaccgcattagggtgagcgaagtcagttttagagctagaaatag-3′)sg57-4(5′-gtggaaaggacgaaacaccggttaatcccactatataaccgttttagagctagaaatag-3′)

Afterwards, the U6-sgRNA fragment was amplified with primers Ins-u6-grna-Fw and -Rv (5′-cgcaattaaccctcactaaagagggcctatttcccatgattcc-3′; 5′-gatgagtttggacaaaccacgcgtcgacagatctcgtctc-3′) and the backbone from pLenti6.3-GFP (provided by GBU, University of Helsinki, Finland) was amplified with primers: BB-Lenti-pcdna-Fw and -Rv (5′-gtggtttgtccaaactcatcaatgtatc-3′; 5′-tttagtgagggttaattgcgcgc-3′). The DNA fragments were fused together using the HiFi assembly master mix (NEB) and transformed in competent bacteria. The generated vectors were analyzed by restriction enzymes and the inserts were verified by Sanger sequencing.

### 2.3. Generation of HEK293-DD-dCas9-VP192_rKSHV.219 Cell Line

To generate HEK293 cells stably expressing DD-dCas9-VP192, the cells were transfected with PB-CAG-DDdCas9VP192-T2A-GFP-IRES-Neo (also deposited in Addgene #102885) together with a vector expressing PiggyBac Transposase as in [21]. The cells were selected in media containing G418 (400 µg/mL). In addition, cells expressing high levels of EGFP, which are also the cells expressing high levels of DD-dCas9-VP192, were sorted using a BD FACSAria II sorter. Prior to sorting, cells were passed through a 40-µm strainer (BD biosciences, Franklin Lakes, NJ, USA) and resuspended in PBS containing 1% fetal calf serum and 2 mM EDTA. Sorted cells were analyzed after expansion by flow cytometry and found mostly GFP-positive (approx. 94%). Next, sorted cells were infected with rKSHV.219 [26] and maintained in media containing G418 (400 µg/mL) and puromycin (5 µg/mL). rKSHV.219 virus production from iSLK.219 and concentration has been described previously [25].

### 2.4. Immunoblotting

Proteins were separated in 4–15% Criterion TGX gels (Bio-Rad, Hercules, CA, USA) and transferred to Trans Blot Turbo Midi Nitrocellulose membranes (Bio-Rad). The membranes were blocked with 5% non-fat dry milk in TBS-T. Antibodies detecting target proteins were diluted in 5% non-fat dry milk in TBS-T and incubated overnight, at 4 °C. Antibodies detecting ORF57 (sc-135746), ORF45 (sc-53883), K8.1 (sc-65446), Actin (sc-8432) were from Santa Cruz Biotechnology (SCBT, Dallas, TX, USA). Antibodies detecting HA tag (HA.11) were from Biolegend (San Diego, CA, USA), Myc-tag (9B11) from Cell Signaling Technology (Danvers, MA, USA) and ORF50 was a gift from Carolina Arias (UC Santa Barbara, Santa Barbara, CA, USA). Anti-Rabbit IgG HRP-linked (7074S) and anti-Mouse IgG HRP-linked (7076S) secondary antibodies were from Cell Signaling Technology. WesternBright Sirius HRP substrate (Advansta, San Jose, CA, USA) was used for revealing the protein bands during imaging in a ChemiDoc gel imaging system (Bio-Rad).

### 2.5. Immunofluorescence Analysis

Cell were fixed in 4% paraformaldehyde or cold methanol, washed with PBS, permeabilized with PBS containing 0.2% Triton-X100 and blocked with PBS containing 0.2% Triton X-100 and 1% bovine serum albumin (BSA). Antibodies detecting HA tag (HA.11) were from Biolegend, anti-ORF50 was a gift from Carolina Arias (UC Santa Barbara, USA) and anti-EGFP was a kind gift from J. Mercer (Institute of Microbiology and Infection, University of Birmingham, UK). Secondary antibodies coupled to Alexa Fluor 488, 555 and 647 were from Thermo Fischer Scientific (Waltham, MA, USA). Nuclei were counterstained with Hoechst 33342 (1 µg/mL, Sigma, St. Louis, MO, USA). Images were acquired using an ImageXpress Pico microscope (Molecular Devices, San Jose, CA, USA), CellInsight High throughput microscope (Thermo Fischer Scientific) or Eclipse Ts2-FL inverted microscope (Nikon, Tokyo, Japan). Images were analyzed using the Cell Profiler 3.0 software package [27] and the following functions to construct the analysis pipelines: Identify Primary Objects, Relate Objects, Filter Objects, Measure Image Intensity, Measure Object Intensity and Export to Spreadsheet.

### 2.6. Virus Titration

rKSHV.219 virus containing supernatants was serially diluted in complete media containing sodium butyrate (1.35 mM, Sigma, St. Louis, MO, USA) and polybrene (8 µg/mL, Sigma). Infection of U2OS cells with serially diluted viruses was achieved by spinoculation (800× *g*, 30 min, RT). Next day, the cells were fixed and stained with an anti-EGFP antibody as described above to identify rKSHV.219-infected cells. Images were acquired using an ImageXpress Pico microscope (Molecular Devices, San Jose, CA, USA) or a CellInsight High throughput microscope (Thermo Fischer Scientific). Images were analyzed using the Cell Profiler 3.0 software package [27] and the following functions were used to construct the analysis pipelines to count infected cells: Identify Primary Objects, Relate Objects, Filter Objects and Export to Spreadsheet.

### 2.7. Quantification of Viral mRNA Transcripts and Genomes

Cellular RNA was extracted using a NucleoSpin RNA extraction kit (Macherey Nagel, Düren, Germany). RNA was reverse transcribed to cDNA using oligo dT primers and Multiscribe reverse transcriptase (Thermo Fischer Scientific). cDNA and target primers were mixed with 2X SYBR (Thermo Fischer Scientific). The genomic DNA was isolated using a Nucleospin Tissue DNA kit (Macherey Nagel, Düren, Germany). The DNA was mixed with 2X SYBR and primers binding to the genomic actin and the viral genome at K8.1. The qPCR reactions were conducted in a Light Cycler 480 qPCR system (Roche, Basel, Switzerland). The primer sequences are listed below:Actin-Fw; Rv (5′-tcacccacactgtgccatctacga-3′; 5′-cagcggaaccgctcattgccaatgg-3′);ORF50-Fw; -Rv (5′-cacaaaaatggcgcaagatga-3′; 5′-tggtagagttgggccttcagtt-3′);ORF45-Fw; Rv (5′-cctcgtcgtctgaaggtga-3′; 5′-gggatgggttagtcaggatg-3′);ORF57-Fw; -Rv (5′-tggacattatgaagggcatccta-3′; 5′-cgggttcggacaattgct-3′);K8.1-Fw; -Rv (5′-aaagcgtccaggccaccacaga-3′; 5′-ggcagaaaatggcacacggttac-3′);Genomic_Actin-Fw; -Rv (5′-agaaaatctggcaccacacc-3′; 5′-aacggcagaagagagaacca-3′).

### 2.8. RNA Sequencing Analysis

RNA Nano 6000 Assay Kit of the Bioanalyzer 2100 system (Agilent Technologies, Santa Clara, CA, USA) was used to determine the RNA integrity. Strand specific poly A enriched mRNA libraries from total cellular RNA were prepared and sequenced in an Illumina Novaseq 6000 (150 bases, paired end) by Novogene (Cambridge, UK). Raw data were cleaned from low quality reads and reads containing adapter and poly-N-sequences in FASTP [28]. Clean reads were mapped to the human genome (GRCh38.p12) using HISAT2 with parameters—dta—phred33 [29]. Sequencing reads that remained unmapped to the human genome were, next, mapped to the KSHV BAC16 [24] (acc: GQ994935), which is identical to rKSHV.219, using STAR [30]. Read counts were generated by FeatureCounts [31]. The PANpromoter-RFP, GFP and Puromycin resistance cassette were added to the GQ994935 sequence to generate the complete rKSHV.219 genome [26]. Raw RNA sequencing data of the viral genes, after removal of human sequences, were deposited at SRA (acc: PRJNA642085).

### 2.9. Statistics

One-Way ANOVA and Student’s t tests were performed in Graph Pad Prism v8.0 (San Diego, CA, USA). For multiple comparison between groups after One-Way ANOVA, Dunnett’s correction was used.

## 3. Results

### 3.1. Targeting the dCas9-Activator to the ORF50 Promoter Induces KSHV Reactivation

We set out to investigate whether a repurposed dCas9 fused to transcription activators can be guided to the KSHV genome to induce the expression of *ORF50* and thereby trigger the complete KSHV lytic replication cycle. For this, we first generated a HEK293 cell line stably expressing a conditionally stabilized form of the dCas9-activator, DD-dCas9-VP192. The VP192 domain consists of 12 copies of the HSV-1 VP16 activator and the destabilizing domain (DD) is derived from the E. coli DHFR protein [32]. This engineered form of DHFR is highly unstable and can be stabilized by trimethoprim (TMP) (schematics in Figure 1A). DD-dCas9-VP192 has been previously used to induce the expression of multiple genes simultaneously to control cell differentiation [21,22]. HEK293 cells, expressing DD-dCas9-VP192, were then stably infected with rKSHV.219, a double-reporter recombinant KSHV virus (rKSHV.219). This virus expresses EGFP from a cellular constitutive promoter (EF1α) and RFP from the PAN viral lytic promoter that is under the control of ORF50 [26]. As a result, EGFP is expressed in all infected cells while RFP is expressed only in cells where the virus is undergoing lytic replication. In the generated cell line, we found that in the absence of TMP, DD-dCas9-VP192 (HA tagged) was localized in both the nucleus and cytoplasm, whereas upon addition of TMP, the stable protein accumulated predominantly in the nucleus. Stabilization and accumulation of the DD-dCas9-VP192 by TMP was confirmed also by immunoblot analysis (Figure 1B).

We transfected the HEK293-DD-dCas9-VP192-rKSHV.219 cells with plasmids expressing sgRNAs binding to different regions upstream of the *ORF50* gene (−300 bp to −50 bp) (Figure 1C). All four guides (sg50-1 to sg50-4) were able to induce the KSHV lytic replication cycle, which was initially detected by the RFP expression (Figure 1C). When compared to non-transfected cells (NT), the mRNA of *ORF50*, *ORF57* (immediate early), *ORF45* (early) and *K8.1* (late) were all upregulated. While sg50-1 and -4 induced lower levels of lytic reactivation with fewer RFP+ cells and lower lytic gene expression, sg50-2 and -3 were more efficient in inducing viral reactivation (Figure 1D). DD-dCas9-VP192-mediated KSHV reactivation also led to virus production, which was further improved upon DD-dCas9-VP192 stabilization by TMP (Figure 1E). The induction of the KSHV reactivation was not fully controllable by TMP for two main reasons. First, because DD-dCas9-VP192 partly retains the ability to activate gene expression also when unstable, as shown previously [21]; and second, when generating the cell line, the cells infected by rKSHV.219 were initially sorted by FACS to express particularly high levels of DD-dCas9-VP192.

Overall, these data demonstrate that the dCas9-fused activators can efficiently induce the expression of ORF50 and the complete lytic cascade leading to virus production.

### 3.2. Transcriptional Activators Fused to dCas9 Can Trigger the Expression of ORF57 without Affecting Viral Latency

Activation of *ORF57* expression in the early phase of the KSHV lytic cycle is dependent on the binding of ORF50 to the promoter of *ORF57* [33]. We next asked whether DD-dCas9-VP192 can activate *ORF57* gene expression without interrupting virus latency. For this, we generated a set of four sgRNA-expressing vectors that target positions -53, -128, -200 and -87 (sg57-1 to -4) upstream of the *ORF57* start codon (ATG) in the KSHV genome (Figure 2A), and tested their efficiency in activating the *ORF57* gene expression. Notably, the sgRNA binding at position -87 bp (sg57-4) was able to induce the strongest expression of *ORF57* (Figure 2B). Next, we compared the effect of sg57-4 to that of sg50-2 on the *ORF50* and *ORF57* gene expression. At mRNA level, there was a slight increase in the basal *ORF50* expression by 2.3 fold in cells expressing *ORF57* only (sg57-4). However, this was a negligible increase when compared to the *ORF50* mRNA levels induced by the ORF50-specific sg50-2 which increased by about 200 fold. *ORF57* expression induced by DD-dCas9-VP192 binding upstream of *ORF57* was about five-fold less compared to that expressed during the lytic cycle induced by sg50-2 (Figure 2C). Accordingly, immunoblot analysis showed high levels of ORF50 and ORF57 proteins upon sg50-2 transfection whereas sg57-4 transfection led to a moderate increase in ORF57 protein levels but no detectable ORF50 protein, as expected (Figure 2D). In addition, while sg50-2 increased viral genome copies by about six fold, indicating the progression of the KSHV lytic replication cycle to the late phase, in the presence of sg57-4, the viral genome copies were unchanged and were comparable to the latent control (Figure 2E). Similarly, the released infectious virus in the supernatant was detected only upon sg50-2, but not upon sg57-4 transfection (Figure 2F). RNA-seq analysis confirmed that binding of DD-dCas9-VP192 upstream of the *ORF57* gene upregulated specifically only the expression of ORF57 without affecting expression of adjacent viral genes whereas binding of DD-dCas9-VP192 upstream of *ORF50* led to a global viral gene expression upregulation as expected during KSHV reactivation (Figure 2G).

This suggests that DD-dCas9-VP192 can be used both to trigger the full KSHV lytic cascade, with increase in intracellular KSHV genomes and release of infectious virus by targeting the *ORF50* promoter and to activate the expression of selected lytic genes (such as *ORF57*) during KSHV latency.

### 3.3. Simultaneous Targeting of the dCas9-Activator at Two Sites on the ORF50 Promoter Further Enhances KSHV Reactivation

It has been shown that binding of the dCas9-activators to two different sites upstream of a cellular gene can further improve the target gene expression [21]. To test this in the context of the viral lytic gene expression, we compared KSHV reactivation in HEK293-DD-dCas9-VP192-rKSHV.219 cells that were transfected with sg50-2 and sg50-3 separately or as a 1:1 mixture of sg50-2 and sg50-3 (sg50-2+3); therefore, total amount of DNA transfected in all conditions (mixture vs. single) was equal. In comparison to sg50-2 or sg50-3 transfected cells, combination of both sgRNAs led to a significant increase in *ORF50*, *ORF57*, *ORF45* and *K8.1* expression at both mRNA and protein levels (Figure 3A,B). Next, we found that viral titers reflected the capacity of sg50-2 and sg50-3 to induce *ORF50* expression. Expression of sg50-2 led to slightly higher virus titers than sg50-3, while combining sg50-2 and -3 (sg50-2+3) improved the viral titers significantly with an almost three-fold increase over the single sgRNAs (Figure 3C). This suggests that targeting DD-dCas9-VP192 at two sites upstream of the *ORF50* gene improves the expression of the KSHV lytic proteins and virus production.

### 3.4. dCas9-Activator Is More Efficient than Ectopic Expression of ORF50 to Induce the Full Viral Lytic Replication Cycle

To compare the KSHV reactivation efficiency in response to targeting of DD-dCas9-VP192 at the *ORF50* promoter to reactivation induced by ectopic *ORF50* expression, we transiently transfected a vector expressing a Myc-tagged *ORF50* or a 1:1 mixture of sg50-2 and sg50-3 vectors (sg50-2+3) in HEK293-DD-dCas9-VP192-rKSHV. At 72 h post transfection, we observed a slightly higher RFP expression in the Myc-ORF50 expressing cells over the sg50-2+3 expressing ones (Figure 4A). Immunoblot analysis showed that while ORF50 protein levels were higher in the Myc-ORF50 transfected cells, the levels of downstream lytic proteins, namely ORF57, ORF45 and K8.1, were higher in the sg50-2+3 transfected cells (Figure 4B). Moreover, the virus titers were four-fold higher in the sg50-2+3-transfected cells than in the Myc-ORF50-overexpressing cells (Figure 4C). At single cell level in the Myc-ORF50-transfected cells, ORF50 was detected in more cells and at higher intensities than in the HEK293-DD-dCas9-VP192-rKSHV-transfected cells with sg50-2+3. This indicates that despite higher levels of ORF50, these cells did not undergo a more efficient, productive KSHV lytic replication (Figure 4D).

Together these results show that ORF50 expression driven by dCas9-mediated gene activation directly from the viral genome can trigger the full KSHV lytic replication program even more efficiently than ectopic expression of ORF50 in terms of viral lytic protein levels and infectious virus production.

## 4. Discussion

The use of dCas9, adapted to modulate cellular gene expression, has revealed high potential in a plethora of applications. Here, we show that dCas9-based activators can be used to efficiently induce ORF50 expression and consequently the full KSHV lytic replication cascade. When compared to ectopic expression of *ORF50*, the DD-dCas9-VP192 induced better both the lytic cycle and virus production. Additionally, we show that besides *ORF50*, dCas9-based activators can be targeted to specifically increase expression of selected KSHV lytic genes, such as *ORF57,* without driving the full lytic gene expression program that culminates with virus production.

The promoter of *ORF50*, a 2.5-kb long region located between the start codons of *ORF45* and *ORF50*, drives the expression of four different *ORF50* isoforms [34]. In this region, there are several RBP-Jk binding sites that serve to potentiate the effect of *ORF50* promoter transactivation by ORF50 itself [35,36,37] through direct interaction of ORF50 with RBP-Jk [33]. Therefore, by directing the binding of the large DD-dCas9-VP192 protein (>250 kb) downstream of the RBP-Jk bindings sites could, in principle, block the RNA polymerase from transcribing the region of *ORF50* downstream of the dCas9 binding site. To our surprise, this did not occur. Not only did KSHV lytic gene expression occur more efficiently but also virus production was not impaired, but rather was enhanced, indicating that dCas9-activators binding to the viral genome do not interfere with either virus replication or packaging.

To date, lymphatic endothelial cells (LECs) are the only reported cell type where the full KSHV lytic replication cycle can occur spontaneously [11,12]. We and others have recently shown that this was partly dependent on the expression of PROX1, which can bind to the ORF50 protein and its promoter to enhance the lytic replication cycle [25,38]. However, in luciferase-based assays, PROX1 alone was unable to enhance the basal activity of ORF50 promoter [25], suggesting that other putative transcription factors expressed in LECs serve to initiate KSHV reactivation in latent cells. Using a dCas9-activator to mimic the function of these putative transcription factors to start ORF50 expression in latent cell, we can orchestrate the lytic replication cascade in a similar fashion as the spontaneous reactivation of KSHV occurs in LECs. Interestingly, when triggered by DD-dCas9-VP192, virus reactivation was even more efficient than when ORF50 was supplied at higher amounts in the same cellular background. Therefore, CRISPRa systems may offer a more physiological alternative to trigger the KSHV lytic cycle as opposed to systems where ORF50 is supplied disproportionally by ectopic expression.

Multiple studies have shown that the ability of dCas9 to bind the DNA is not hampered by fusing it to other functional domains at either ends of the protein. In fact, in our experiments we used a dCas9 fused to a destabilizing domain (based on *E. coli* DHFR) at the N-terminus and an activator domain (VP192) at the C-terminus to induce ORF50 expression and initiate the viral lytic replication cycle [19,20,21,22,23,39]. It is conceivable that a dCas9 fused to the activator domain of ORF50 would drive a more efficient viral reactivation than the DD-dCas9-VP192 used in this study by recruiting (at the *ORF50* promoter) authentic ORF50 interaction partners. The ORF50 activator domain is at the C-terminus of the protein between the amino acids 486-691. This part of the protein binds also to K8 (KbZIP), an early KSHV protein and transcription factor, which has been shown to modulate KSHV reactivation [40]. Although a KSHV ORF50 transactivation domain fused to dCas9 remains to be tested, a recent study used a dCas9 fused to the Epstein-Barr virus (EBV) RTA (highly homologous to KSHV ORF50) together with VP64 (4 × VP16) and p65 to achieve potent expression of the target genes [39].

In summary, we show that dCas9-activator can be used to induce the full KSHV lytic replication cascade by directing its binding upstream of the *ORF50* gene. In addition, the KSHV-infected HEK293 cell model expressing a dCas9-activator offers a versatile screening platform to identify and validate new regulators of the KSHV biology. In the future, CRISPRa technology can be applied also to modulate KSHV gene expression in physiologically relevant KSHV-infected cell models such as PEL-derived cell lines or human endothelial cells.

## Figures and Tables

**Figure 1 viruses-12-00952-f001:**
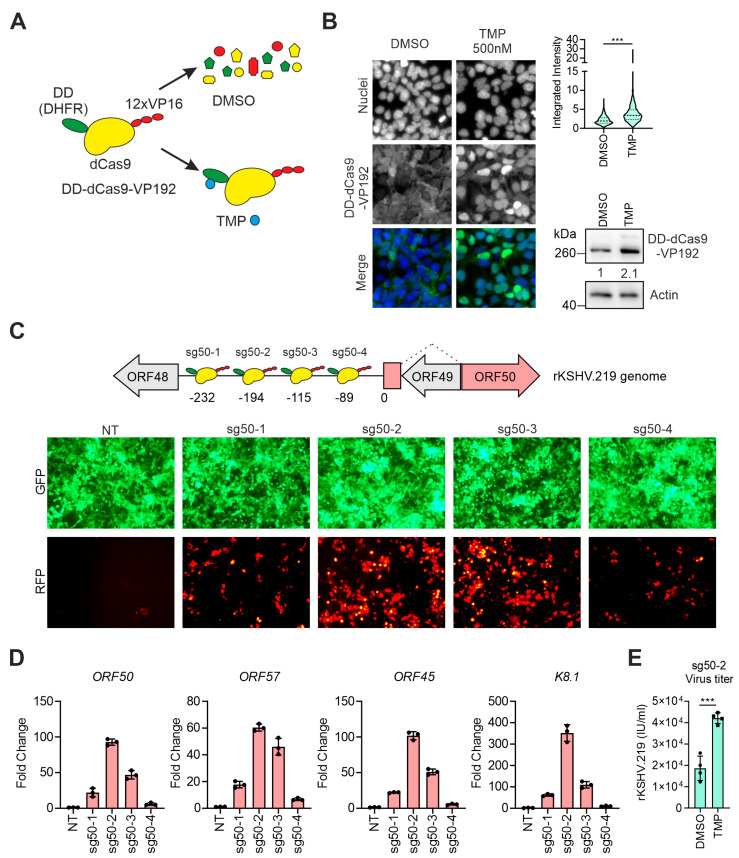
Targeting a nuclease-deficient Cas9 (dCas9) based activator to the *ORF50* promoter induces Kaposi’s sarcoma-associated herpesvirus (KSHV) reactivation. (**A**) Schematic representation of destabilizing domain (DD)-dCas9-VP16 activation domain (VP192) stabilization upon addition of trimethoprim (TMP). (DHFR: DD–dihydrofolate reductase-derived destabilization domain, VP192–12X VP16) (**B**) DD-dCas9-VP192 localization in HEK293-DD-dCas9-VP192_rKSHV.219 treated with DMSO or TMP (500 nM) for 72 h. Intracellular localization of DD-Cas9-VP192 was detected by anti-HA antibody staining and nuclei were counterstained with Hoechst 33342. Representative images (left) and quantification of DD-Cas9-VP192 fluorescence signal (violin plot, right) in the area occupied by each individual nucleus (*n* > 500, Student’s t test, *** −*p* < 0.001). Below the violin plot, immunoblot analysis for the indicated proteins of HEK293-DD-dCas9-VP192-rKSHV.219 cells treated with DMSO or TMP (500 nM) for 72 h. Actin normalized band quantification is shown for DD-dCas9-VP192, below the bands. (**C**) Upper panel: schematic representation of DD-dCas9-VP192 binding sites upstream of *ORF50* upon expression of the indicated single-guide RNAs (sgRNAs). Numbers indicate nucleotides upstream of *ORF50* start codon (indicated as 0). Lower panels: GFP and RFP expression in HEK293_DD-dCas9-VP192_rKSHV.219 non-transfected (NT) or transfected with vectors expressing the indicated sgRNA and treated with TMP (500 nM) for 72 h. RFP is a marker of the rKSHV.219 lytic replication cycle. (**D**) Quantification of *ORF50, ORF57, ORF45* and *K8.1* mRNA levels in cells treated as in (**C**). Each dot represents a biological replicate, bars show average, error bars show SD. (**E**) Quantification of rKSHV.219 titers in the supernatant of HEK293-DD-dCas9-VP192-rKSHV.219 cells transfected with sg50-2-expressing plasmid and treated with DMSO or TMP (500 nM) for 72 h. Each dot represents a biological replicate, bars show average, error bars show SD, Student’s t test, *** −*p* < 0.001.

**Figure 2 viruses-12-00952-f002:**
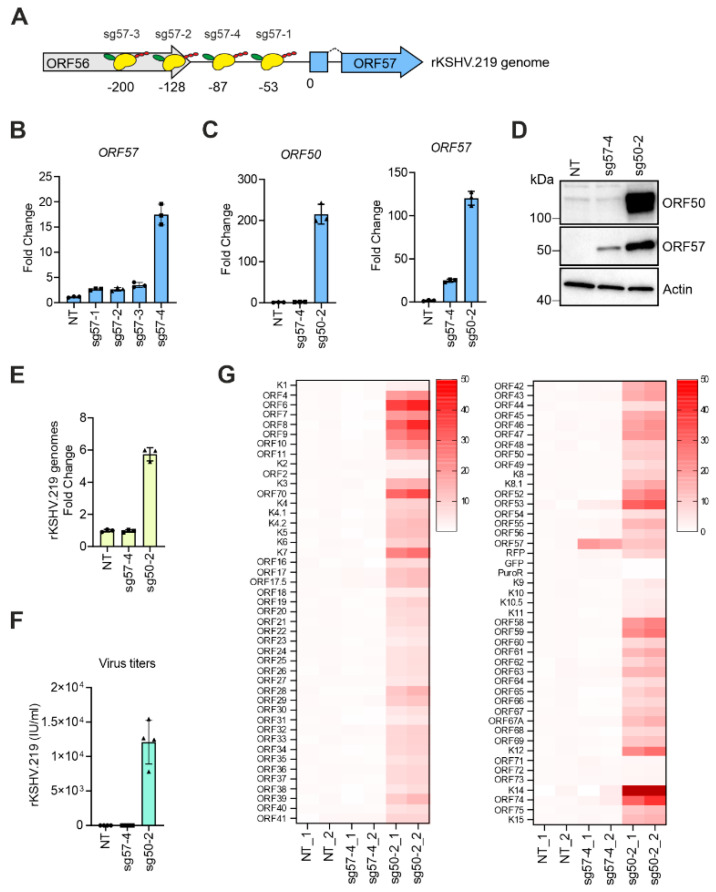
Activators fused to dCas9 can trigger the expression of *ORF57* without disrupting the viral latency. (**A**) A schematic representation of DD-dCas9-VP192 binding sites upstream of the *ORF57*gene upon expression of the indicated sgRNAs. Numbers indicate nucleotides upstream of *ORF57* start codon (ATG), shown as 0. (**B**–**G**) HEK293_DD-dCas9-VP192_rKSHV.219 cells were transfected with vectors expressing the indicated sgRNA or left non-transfected (NT) and treated with TMP (500 nM) for 72 h. (**B**) Analysis of *ORF57* mRNA levels. (**C**) Analysis of *ORF50* and *ORF57* mRNA levels. (**D**) Immunoblot analysis for the indicated proteins. (**E**) Quantification of the intracellular KSHV genome copies. (**F**) Quantification of rKSHV.219 titers in the supernatant. (**G**) Heatmap of KSHV gene expression after RNA-seq analysis of samples in (**C**) (*n* = 2 biological replicates). The read counts were normalized to NT samples and to genome copy numbers calculated in (**E**). Dark red tiles show values above 50. (**B**,**C**,**E**,**F**) Each dot represents a biological replicate, bars show average, error bars show SD.

**Figure 3 viruses-12-00952-f003:**
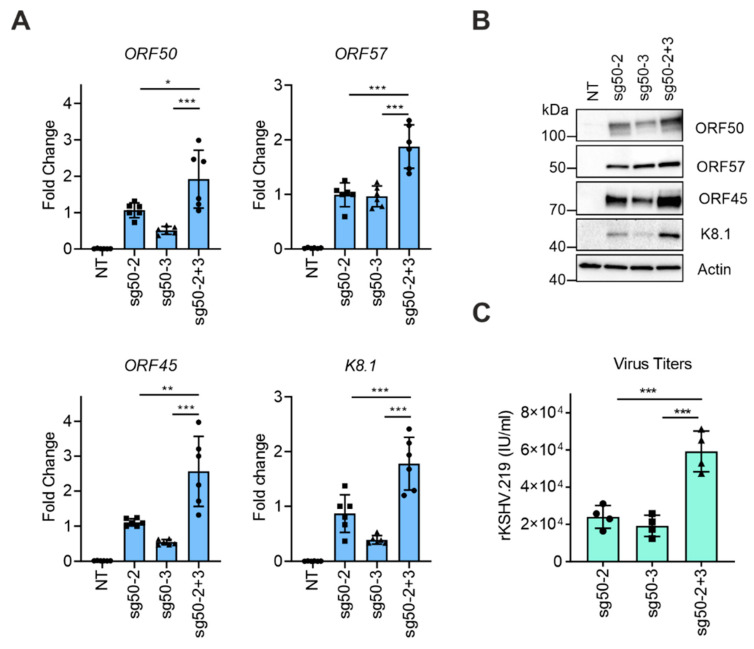
Simultaneous targeting of the dCas9-activator at two sites on the *ORF50* promoter enhances KSHV reactivation. (**A**–**C**) HEK293_DD-dCas9-VP192_rKSHV.219 cells were transfected with vectors expressing the sgRNA sg50-2, sg50-3, both (sg50-2+3) or left non-transfected (NT) and treated with TMP (500 nM) for 72 h. (**A**) Quantification of *ORF50*, *ORF57*, *ORF45* and *K8.1* mRNA levels. (**B**) Immunoblot analysis for the indicated proteins. (**C**) Quantification of rKSHV.219 titers in the supernatant. (**A** and **C**) Each dot represents a biological replicate, bars show average, error bars show SD. One-Way ANOVA followed by Dunnett’s correction for multiple comparisons testing were used to calculate whether differences between sg50-2, sg50-3 and sg50-2+3 groups were significant (*** −*p* < 0.001, ** −*p* < 0.01, * −*p* < 0.05).

**Figure 4 viruses-12-00952-f004:**
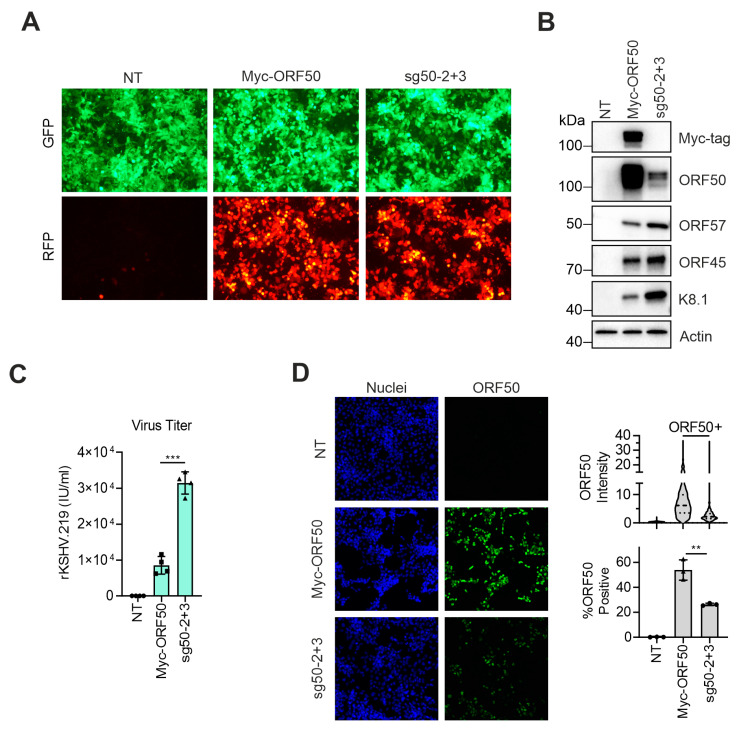
dCas9-activator is more efficient than ectopic expression of *ORF50* to induce the viral lytic replication. (**A**–**D**) HEK293_DD-dCas9-VP192_rKSHV.219 cells were transfected with vectors expressing the sgRNA sg50-2 and sg50-3 (sg50-2+3) or Myc-ORF50 and treated with TMP (500 nM) for 72 h. (**A**) Representative images showing GFP (upper row) and RFP (lower row) expression of cells transfected with sg50-2 and sg50-3 (sg50-2+3), Myc-ORF50 or non-transfected (NT). RFP is a marker of the rKSHV.219 lytic replication cycle. (**B**) Immunoblot analysis for the indicated proteins. (**C**) Quantification of rKSHV.219 titers in the supernatant. (**D**) Quantification of ORF50 expression after staining of the cells with an anti-ORF50 antibody. Nuclei were counterstained with Hoechst 33342. (**C**,**D**) In the violin plot, *n* > 450 nuclei/condition. In the bar graphs, each dot represents a biological replicate, bars show average and error bars show SD. Student’s t test was used to calculate whether differences between Myc-ORF50 and sg50-2+3 groups were significant (** −*p* < 0.01, *** −*p* < 0.001).

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
