# Peer review of "Kaposi’s Sarcoma-Associated Herpesvirus Reactivation by Targeting of a dCas9-Based Transcription Activator to the ORF50 Promoter"

_viruses, 2020, doi:10.3390/v12090952_

Round 1

Reviewer 1 Report

Summary:

KSHV is associated with Kaposi sarcoma, multicentric Castleman disease, primary effusion lymphoma, and KSHV inflammatory cytokine syndrome. While KSHV is predominantly detected as a latent infection in these associated malignancies, lytic replication of KSHV is considered an important aspect of pathogenesis. Regulation of lytic reactivation is typically investigated under the induction with non-physiological chemical agents. To better model physiological induction of KSHV lytic reactivation, the authors use a novel system to regulate the lytic switch transactivator RTA. They report that: 1) expression of KSHV latency to lytic switch protein RTA and full cascade of KSHV lytic replication was specifically induced in latently infected cell by dCas9-VP192 recruited to RTA promoter region; 2) they could apply a similar strategy to trigger expression of an RTA-regulated viral gene, ORF57 while bypassing expression of RTA; 3) two sgRNAs targeting different regions of RTA promoter led to a synergistic effect; 4) KSHV lytic reactivation was more potently induced upon introduction of dCas9-VP192/sgRNA complex targeting RNA promoter than overexpression of an RTA protein. Overall, barring a few technical concerns, this is a novel technology that might inform our understanding of the molecular events that drive KSHV reactivation and represent a useful tool for cell culture studies.

Text revisions would address most concerns, but a demonstration that guideRNA expression levels (or transfection efficiency) are comparable and that RTA is regulatable by TMP in the context of infection for one critical experiment (perhaps sg502+3) are needed to confirm this technology would be useful for the field.

Major comments:

  1. Technical concerns.
  • The authors are comparing guideRNAs that target the regulatory regions of ORF50 and ORF57. However, there are indications of variation in transfection efficiency when comparing replicate experiments (Figure 2F vs Figure 3F). Making conclusions regarding efficacy of different gRNAs would be bolstered by experiments that demonstrate similar transfection efficiencies or levels of gRNAs.
  • The dCas9 fusion protein is an updated version of first generation of dCas9 activation system (dCas9-VP16) that generates 10-20 fold activation of target genes. The activation of ORF50 showed in Figure 1D is moderate. A comparison to other versions of dCas9 activation protein in the literature such as dCas9-VPR (VP16-p65-RTA) could be compared in parallel to this newest construct to confirm this VP192 system is an improvement in technology.
  • There is no analysis comparing RTA levels with and without TMP. The regulatable degradation/stability of this system is novel, but it was never evaluated in the context of infection.
  • How was the specificity of designed sgRNAs validated? There is an intriguing result that shows dCas9-VP192/sg50 2+3 induced less RTA than exogenous overexpressed RTA protein, yet resulted in a higher level of downstream lytic protein and released virions. Given that the lytic cascade is strictly dominated by the expression of RTA, one simple explanation would be that there was unspecific binding of sg50 2+3 to downstream lytic genes or cellular gene loci that facilitate KSHV lytic reactivation. Please address this potential issue.
  1. Statistical analysis and other methods
  • What was the scale for experiments measuring viral titer in sups? The yields in Figure 2F and Figure 3F seemed pretty low considering 293-rKSHV.219 cells are used for production.
  • There is a lack of statistical details: error bars are missing, replicates and statistical tests are not described for all subpanels in every figure.
  • The normalization by genome loads is curious since an increase in genome load would be expected to accompany full lytic gene expression (thus lytic gene expression is underreported). The RNAseq section does not describe the computational pipeline.
  1. Implications for the field:
    • How was the specificity of designed sgRNAs validated? There is an intriguing result that shows dCas9-VP192/sg50 2+3 induced less RTA than exogenous overexpressed RTA protein, yet the dCas9a system resulted in a higher level of downstream lytic protein and released virions. Given that the lytic cascade is strictly dominated by the expression of RTA, one simple explanation would be that there was unspecific binding of sg50 2+3 to downstream lytic genes or cellular gene loci that facilitate KSHV lytic reactivation. Please address this potential issue.
    • Lines 313-315, expression of lytic genes is technically not compatible with a latency program. In addition, other lytic genes were expressed with ORF57 induction. Perhaps another interpretation is that ORF57 alone was not sufficient to drive full lytic gene expression that culminates in infectious particle production?
    • Lines 326-328. This can not be stated if the host transcription factors were not evaluated.

Minor comments:

  1. Line 1, Article
  2. Line 12, which authors are associated with institute 5?
  3. Line 189, it should be ‘EF1α’.
  4. The relative locations of sgRNAs should be consistent- is this the start of transcription or translation (sgRNAs for 57 in Figure 2A)?
  5. Figure 1, Define NT
  6. MW markers are absent in western blots
  7. Figure 4D, The violin plot with a broken axis is a bit unusual and unnecessary.
  8. Lines 332-335, The meaning of this sentence is unclear, rewording is needed.

Author Response

Reviewer #1

Summary:

KSHV is associated with Kaposi sarcoma, multicentric Castleman disease, primary effusion lymphoma, and KSHV inflammatory cytokine syndrome. While KSHV is predominantly detected as a latent infection in these associated malignancies, lytic replication of KSHV is considered an important aspect of pathogenesis. Regulation of lytic reactivation is typically investigated under the induction with non-physiological chemical agents. To better model physiological induction of KSHV lytic reactivation, the authors use a novel system to regulate the lytic switch transactivator RTA. They report that: 1) expression of KSHV latency to lytic switch protein RTA and full cascade of KSHV lytic replication was specifically induced in latently infected cell by dCas9-VP192 recruited to RTA promoter region; 2) they could apply a similar strategy to trigger expression of an RTA-regulated viral gene, ORF57 while bypassing expression of RTA; 3) two sgRNAs targeting different regions of RTA promoter led to a synergistic effect; 4) KSHV lytic reactivation was more potently induced upon introduction of dCas9-VP192/sgRNA complex targeting RNA promoter than overexpression of an RTA protein. Overall, barring a few technical concerns, this is a novel technology that might inform our understanding of the molecular events that drive KSHV reactivation and represent a useful tool for cell culture studies.

Text revisions would address most concerns, but a demonstration that guideRNA expression levels (or transfection efficiency) are comparable and that RTA is regulatable by TMP in the context of infection for one critical experiment (perhaps sg502+3) are needed to confirm this technology would be useful for the field.

Response: We thank the reviewer for the comments and for appreciating the novelty and utility of this system in the field. A point by point response is provided below.

Major comments:

Technical concerns.

Point 1: The authors are comparing guideRNAs that target the regulatory regions of ORF50 and ORF57. However, there are indications of variation in transfection efficiency when comparing replicate experiments (Figure 2F vs Figure 3F). Making conclusions regarding efficacy of different gRNAs would be bolstered by experiments that demonstrate similar transfection efficiencies or levels of gRNAs.

Response to point 1: We thank the reviewer for pointing out this issue. RT-qPCR analysis shows that ORF50 mRNA increases consistently in both Fig 2C and old Fig 3 (A-D) experiments by about 200 times. Therefore, the differences in virus titer between fig 2F and fig 3F are not due to variations in the transfection efficiency or differential expression of sgRNAs but rather variations relating to virus titer quantification. These could include problems with the polybrene batch that was used to improve rKSHV.219 infection conditions of U2OS or with the GFP antibody used to enhance the GFP signal from infected cells in order to facilitate quantification after imaging in our high throughput microscopes. We have therefore repeated the experiment in fig 2F once more with a new batch of polybrene and GFP antibody. The titers improved and the new results are now shown in Fig 2F of the revised manuscript.

Point 2: The dCas9 fusion protein is an updated version of first generation of dCas9 activation system (dCas9-VP16) that generates 10-20 fold activation of target genes. The activation of ORF50 showed in Figure 1D is moderate. A comparison to other versions of dCas9 activation protein in the literature such as dCas9-VPR (VP16-p65-RTA) could be compared in parallel to this newest construct to confirm this VP192 system is an improvement in technology.

Response to point 2: Considering that the KSHV chromatin structure in latently infected cells is tightly repressed and thus, inaccessible to transcription activators in general, we were quite satisfied to get a substantial upregulation of RTA and MTA (ORF57) transcription (100-fold) and increase in virus production. In this study, our goal was to provide a proof-of-principle that CRISPR activation technology can be applied to modify viral gene expression. However, while it is a great idea to compare a dCas9 fused to possibly better activators derived from KSHV encoded transcription factors in a more physiological cellular background (e.g. in PEL cell lines), it will be the focus of a future study.

Point 3: There is no analysis comparing RTA levels with and without TMP. The regulatable degradation/stability of this system is novel, but it was never evaluated in the context of infection.

Response to point 3: Yes, stabilization of DD-dCas9-VP192 leads to better virus production (thus also reactivation). We have added into Fig 1 panel E a graph showing that rKSHV.219 titers increase upon addition of TMP. However, reactivation by DD-dCas9-VP192 was not fully tunable for two main reasons. First, prior to infection with rKSHV.219, we collected by FACS cells that expressed highest level of DD-dCas9-VP192. This is already mentioned in the Materials and methods section 2.3. Generation of HEK293-DD-dCas9-VP192-rKSHV.219 cell line. And second, as previously shown (PMID: 26352799), the unstable form of DD-dCas9-VP192 (in the absence of TMP) still retains the ability to induce gene expression but to a lesser extent than the TMP stabilized form. Moreover, in the same publication (PMID: 26352799), Balboa et al. show that expression of dCa9-VP192 (without DHFR) by a doxycycline inducible promoter is also leaky. Hence, to generate a fully drug inducible dCas9-activator system Balboa et al. had to combine both systems, which is, expression of DD-dCas9-VP192 from a dox inducible promoter. We avoided this because DD-dCas9-VP192 did not appear to affect the KSHV latency and to keep our system as simple as possible. Moreover, our aim was only to show that CRISPR activation technology can be applied to modulate viral gene expression, as mentioned above.

Point 4: How was the specificity of designed sgRNAs validated? There is an intriguing result that shows dCas9-VP192/sg50 2+3 induced less RTA than exogenous overexpressed RTA protein, yet resulted in a higher level of downstream lytic protein and released virions. Given that the lytic cascade is strictly dominated by the expression of RTA, one simple explanation would be that there was unspecific binding of sg50 2+3 to downstream lytic genes or cellular gene loci that facilitate KSHV lytic reactivation. Please address this potential issue.

Response to point 4: We thank the reviewer for raising this issue. The sgRNA were designed using the tools in the crispr.mit.edu website. Among the highest scoring sgRNA (with least off-target effects), we chose only those four that were binding to different sites in the viral genome.

To address the following statement of this Reviewer: “dCas9-VP192/sg50 2+3 induced less RTA than exogenous overexpressed RTA protein” when repeating the experiment in Figure 3 (requested by reviewer 3), we compared also how ORF50 and K8.1 were upregulated when KSHV reactivation was triggered by sg50-2 and sg50-3 alone or in combination to the condition when Myc-ORF50 was ectopically expressed. Please note that Myc-ORF50 transcripts are not detected by RT-qPCR as the Fw primer binds only partially to these transcripts as the other half binds to the untranslated region of the ORF50 in the KSHV genome. The annealing temperature of ORF50-Fw primer to Myc-ORF50 transcripts is Ta = 39°C which is far below the annealing temperature we set in our RT-qPCR program, Ta = 60°C. Therefore, in all conditions only the ORF50 mRNA expressed by the KSHV genome is measured.

We found that Myc-ORF50, although abundantly expressed (Figure 4), activated ORF50 expression and the downstream late lytic K8.1 gene expression comparably to sg50-2. Whereas, the sg50-2+3 combination improved ORF50 and K8.1 expression in a synergistic manner. Therefore, it is unlikely that the better lytic reactivation obtained with sg50-2+3 is the result of an unspecific effect of either sgRNAs, since alone, they induced a similar KSHV reactivation to myc-ORF50 overexpression.

Figure 2. HEK293-DD-dCas9-VP192-rKSHV.219 were transfected with vectors expressing the indicated sgRNAs, or Myc-ORF50 and treated with TMP (500 nM) for 72 h. Quantification of ORF50, and K8.1 transcripts by RT-qPCR.

Statistical analysis and other methods

Point 5: What was the scale for experiments measuring viral titer in sups? The yields in Figure 2F and Figure 3F seemed pretty low considering 293-rKSHV.219 cells are used for production.

Response to point 5:  We thank the reviewer for bringing up this important point. In our study, virus titer quantification was done at 72 h post-transfection since at this time point, the non-transfected (NT) cells became confluent. We have not directly compared virus yields of our cell line transfected with sgRNAs targeting ORF50 promoter to the 293-rKSHV.219 cell line. However, in our hands, when compared to iSLK.219, which we routinely use for rKSHV.219 production, the virus yields are about 3-4-fold lower. Titers in iSLK.219 induced with dox for 72h are about 2X10e5 IU/ml as we recently published in (PMID: 32518203 supplementary figure 4F) and in Fig 3F of this manuscript about 6x10e4.  Nevertheless, we are not proposing to replace the current cell lines for production of high virus titer stocks with the cell line generated in this study but rather characterizing here a new tool for the field.

Point 6: There is a lack of statistical details: error bars are missing, replicates and statistical tests are not described for all subpanels in every figure.

Response to point 6: We have repeated the experiments in Fig 1 C-D and Fig 2A using more biological replicates. Statistical tests are now included in all figure legends.

Point 7: The normalization by genome loads is curious since an increase in genome load would be expected to accompany full lytic gene expression (thus lytic gene expression is underreported). The RNAseq section does not describe the computational pipeline.

Response to point 7: We agree that the lytic gene expression is underreported for the samples of sg50-2. However, to better appreciate the viral gene expression from each individual viral genome under the conditions when ORF50 and ORF57 were induced by sg50-2 and sg57-4, we decided to normalize to the viral genome copies quantified in Fig 2E. We think this is a fairer way to show our data, especially when comparing the two very different situations; 1) when only ORF57 is expressed (sg57-4) to 2) when ORF50, inducing the full lytic cycle, is expressed (sg50-2).

We have added more information in the ¨Materials and methods section of RNA-seq to studies describing further the computational pipelines.

Implications for the field:

Point 7: How was the specificity of designed sgRNAs validated? There is an intriguing result that shows dCas9-VP192/sg50 2+3 induced less RTA than exogenous overexpressed RTA protein, yet the dCas9a system resulted in a higher level of downstream lytic protein and released virions. Given that the lytic cascade is strictly dominated by the expression of RTA, one simple explanation would be that there was unspecific binding of sg50 2+3 to downstream lytic genes or cellular gene loci that facilitate KSHV lytic reactivation. Please address this potential issue.

Response point 7: This is a repetition of Point 4 which was addressed above.

Point 8: Lines 313-315, expression of lytic genes is technically not compatible with a latency program. In addition, other lytic genes were expressed with ORF57 induction. Perhaps another interpretation is that ORF57 alone was not sufficient to drive full lytic gene expression that culminates in infectious particle production?

Response to point 8: We thank the reviewer for the suggestion. We have now changed the sentence to (line 345-347 in the revised text): “Additionally, we show that besides ORF50, dCas9 based activators can be targeted to specifically increase expression of selected KSHV lytic genes, such as ORF57, without driving the full lytic gene expression that culminates with virus production.”  

Lines 326-328. This can not be stated if the host transcription factors were not evaluated.

Response: We have removed this sentence.

Minor comments:

Line 1, Article

Response: Now line 1 shows Article.

Line 12, which authors are associated with institute 5?

Response: This was a mistake, and the numbering for the 1-4 of author affiliations is now corrected.

Line 189, it should be ‘EF1α’.

Response: Line 201 in the revised manuscript now has the correct name for ‘EF1α’.

The relative locations of sgRNAs should be consistent- is this the start of transcription or translation (sgRNAs for 57 in Figure 2A)?

Response: As indicated in the Figure 1 and 2 legends “ORF57/ORF50 start codon”, which is the start of translation.

Figure 1, Define NT

Response: NT is now defined in the Figure 1 legend, line 220.

MW markers are absent in western blots

Response: Western blots now show molecular weights in the revised version.

Figure 4D, The violin plot with a broken axis is a bit unusual and unnecessary.

Response: We agree but as the outlier values make the violin plots look flat, we would prefer to leave the plots as they are, to better appreciate the differences between the samples in the lower ranges.

Lines 332-335, The meaning of this sentence is unclear, rewording is needed.

Response: Upon request by the Reviewer 3, we have modified the discussion further and this sentence was removed.

Reviewer 2 Report

The study provides an interesting alternative method of reactivation for KSHV lytic replication. Notably, it offers the opportunity to activate specific genes within the KSHV genome independently of the lytic cycle and, importantly, allows reactivation by directly focusing the ORF50 promoter.

General comments

  • Presumably, dCas9 could be used to directly activate other KSHV lytic genes (as suggested by the authors in their targeting of ORF57). I feel it is a shame that the authors did not investigate this with proteins not in the immediate-early phase of the life cycle e.g. ORF65, ORF47. I feel this is an important control to perform for the authors to be able to confidently state that selected KSHV genes can be reactivated without interrupting latency.
  • To fully demonstrate the effectiveness of their dCas9 method the authors should have used existing strategies (e.g. TPA/sodium butyrate or TREx BCBL-Rta) as controls within their experiments. I would have found it a more interesting study to see this comparison with existing strategies. The over-expression of myc-RTA in figure 4 was a good example of how this approach could have improved the study, as this was an interesting experiment, but would have been of more interest to the wider community had this comparison been performed on more commonly used systems.
  • Whilst it may be beyond the scope of the experimental procedures in this instance, a more relevant cell line throughout would further emphasise the usefulness of this method. HEK293 cells are routinely used for KSHV research because they are easy to work with and KSHV can be reactivated within them. However, the are not physiologically relevant in terms of KSHVs natural lifecycle. Given the flexibility of a CRISPR system, I would like to see the authors discuss the options for more relevant cell lines e.g. B cells, HUVECS.
  • The authors mention that the HSV VP16 protein is used as an activator because of evidence of it's function in the literature, and that other literature suggests relevant viral activators are extremely potent (their EBV example), and the authors elude to RTA having the potential to offer very high lytic reactivation levels. I am slightly disappointed that the authors did not investigate the effect of using the RTA activator domain experimentally, and instead chose only to discuss the possibility. This would have enhanced the study.

Specific comments

Lines 2, 24, 40: more usual nomenclature is "Kaposi's sarcoma-associated herpesvirus". Suggest changing to this.

Line 116: the authors should give details of the cell sorting performed.

Line 192: I found the staining in Figure 1B a little confusing. As these are rKSHV.219 cells, I expect to see all cells expressing GFP. The authors choice of pseudocolouring their anti-HA green is confusing given the expectance to see GFP in these cells - this is assuming the authors used a secondary antibody other than AlexaFluor488 (although this is not stated), and which I would hope they didn't use.

Lines 216-218: I'm not sure the reason to investigate whether expression of ORF57 can induce the lytic cycle is justified, given that it is well documented that ORF57 is not sufficient on its own to reactivate the lytic cycle. I suggest the authors re-word to highlight the opportunity of their system to investigate expression of KSHV proteins without interrupting latency.

Lines 223-225: was this a statistically significant increase in ORF50 mRNA levels?

Line 232: the word "also" not required.

Line 297: Figure 4D, the immunofluorescence image is a little hard to make out. I suggest adjusting the contrast/brightness to make the image clearer being careful not alter the data.

Author Response

Reviewer #2

The study provides an interesting alternative method of reactivation for KSHV lytic replication. Notably, it offers the opportunity to activate specific genes within the KSHV genome independently of the lytic cycle and, importantly, allows reactivation by directly focusing the ORF50 promoter.

Response: We thank the reviewer for finding our work relevant and recognizing the novelty of the study. A point by point response is provided below.

General comments

Point 1: Presumably, dCas9 could be used to directly activate other KSHV lytic genes (as suggested by the authors in their targeting of ORF57). I feel it is a shame that the authors did not investigate this with proteins not in the immediate-early phase of the life cycle e.g. ORF65, ORF47. I feel this is an important control to perform for the authors to be able to confidently state that selected KSHV genes can be reactivated without interrupting latency.

Response to point 1: We chose to induce the immediate-early ORF57 viral gene because ORF57 protein plays a role in the stabilization and export of the viral transcripts. Therefore, we considered it as a suitable candidate to test whether latency would be affected by DD-dCas9-VP192 induction of a viral gene. However, using DD-dCas9-VP192, we have activated also the expression of ORF36 and ORF21, which according to PMID: 24453964, are not immediate early genes. ORF21 and ORF36 are expressed at 24-48 h and 48-72 h, respectively, post induction of iSLK.219 with doxycycline.  Notably, also in this case there was no perturbation of the viral latency when either ORF36 or ORF21 expression were induced. These data will be part of a future study that we are still working on.

Point 2: To fully demonstrate the effectiveness of their dCas9 method the authors should have used existing strategies (e.g. TPA/sodium butyrate or TREx BCBL-Rta) as controls within their experiments. I would have found it a more interesting study to see this comparison with existing strategies. The over-expression of myc-RTA in figure 4 was a good example of how this approach could have improved the study, as this was an interesting experiment, but would have been of more interest to the wider community had this comparison been performed on more commonly used systems.

Response to point 2: BCBL1-TREX-RTA and iSLK.219 are great models for studying KSHV reactivation and production of high titer virus stocks. In this study, we are not suggesting to replace the current, existing systems but rather complementing them with the CRISPRa system described here. Therefore, we did not perform the suggested comparisons and instead decided to compare side by side exogenous overexpression of a Myc-RTA to the combination of sg50-2 and sg50-3 within the same cellular background.

Point 3: Whilst it may be beyond the scope of the experimental procedures in this instance, a more relevant cell line throughout would further emphasise the usefulness of this method. HEK293 cells are routinely used for KSHV research because they are easy to work with and KSHV can be reactivated within them. However, the are not physiologically relevant in terms of KSHVs natural lifecycle. Given the flexibility of a CRISPR system, I would like to see the authors discuss the options for more relevant cell lines e.g. B cells, HUVECS.

Response to point 3: Thank you for the suggestion. We have added in the last paragraph of Discussion a section discussing this perspective (Line 386-388).

Point 4: The authors mention that the HSV VP16 protein is used as an activator because of evidence of its function in the literature, and that other literature suggests relevant viral activators are extremely potent (their EBV example), and the authors elude to RTA having the potential to offer very high lytic reactivation levels. I am slightly disappointed that the authors did not investigate the effect of using the RTA activator domain experimentally, and instead chose only to discuss the possibility. This would have enhanced the study.

Response to point 4: In this particular study, we wanted to use an activator domain that would not affect KSHV latency upon ectopic expression. We chose VP16 as both iSLK.219 and TREx-BCBL1-RTA express an rtTA fused to multiple copies of VP16. It is actually in our future plans to optimize a dCas9 protein fused the KSHV RTA activation domain that can efficiently activate not only KSHV genes but also cellular genes.

Specific comments

Lines 2, 24, 40: more usual nomenclature is "Kaposi's sarcoma-associated herpesvirus". Suggest changing to this.

Response: Virus nomenclature in line 2, 24 and 40 was changed as suggested.

Line 116: the authors should give details of the cell sorting performed.

Response: We have provided more details on the cell sorting in the Materials and methods section 2.3. Generation of HEK293-DD-dCas9-VP192rKSHV.219 cell line (see lines 114-119).

Line 192: I found the staining in Figure 1B a little confusing. As these are rKSHV.219 cells, I expect to see all cells expressing GFP. The authors choice of pseudocolouring their anti-HA green is confusing given the expectance to see GFP in these cells - this is assuming the authors used a secondary antibody other than AlexaFluor488 (although this is not stated), and which I would hope they didn't use.

Response: These cells were fixed and permeabilized in ice-cold methanol. This fixation method quenches the GFP signal originating from rKSHV.219. But indeed, the anti-mouse secondary antibody used for detection of the HA-tag antibody (mouse) was coupled to Alexa-647 (far-red). Green is just a pseudocolor.

Lines 216-218: I'm not sure the reason to investigate whether expression of ORF57 can induce the lytic cycle is justified, given that it is well documented that ORF57 is not sufficient on its own to reactivate the lytic cycle. I suggest the authors re-word to highlight the opportunity of their system to investigate expression of KSHV proteins without interrupting latency.

Response: Thank you for the suggestion. We now reworded the second sentence of Section 3.2 “We next asked whether DD-dCas9-VP192 can activate ORF57 gene expression without interrupting virus latency.”

Lines 223-225: was this a statistically significant increase in ORF50 mRNA levels?

Response: Yes, it is a significant increase (Student’s t test, p=0.027). However, as we did not detect ORF50 protein band by immunoblot nor viruses in the supernatants of cells transfected with sg57-4, we therefore concluded that ORF57 induction by DD-dCas9-VP192 did not perturb latency and that the ORF50 mRNA increase by 2.3-fold is negligible as stated in lines 253-256.

Line 232: the word "also" not required.

Response: “also” was removed from the sentence (line 261 in revised manuscript)

Line 297: Figure 4D, the immunofluorescence image is a little hard to make out. I suggest adjusting the contrast/brightness to make the image clearer being careful not alter the data.

Response: Thank you for the suggestion. We have improved the contrast/brightness equally across all images in 4D. We want to point out, however, that quantification of fluorescence in the images was done using the original, unaltered Tif bio-format images.

Reviewer 3 Report

In this study, the authors use a previously designed CRISPRa system to trigger KSHV reactivation. They show that this system works well to either target individual viral genes or trigger the whole reactivation cascade by inducing ORF50 expression. To my knowledge, it is the first time that someone tried to use CRISPRa on the KSHV genome. This is a potentially important new tool that could be of interest to the KSHV community. However, to better support their claims, the authors should address the following points:

- The authors use TMP to send their dCas9 construct to the nucleus where it will presumably act on the viral genome. It is unclear from the manuscript but I would assume that the cell line they further create (combining dCas9 expression and the KSHV genome) would have to be under constant TMP condition? If so, the authors should include a control where they use TMP alone (no sgRNA) and see the effect on viral gene expression.

- Still on TMP: In the articles cited (Ref 21 and 22), TMP is used to increase stability of the construct but here the author show that it’s the localization that is impacted by TMP (Fig 1B) – how do the authors explain this difference? Do the authors see any difference in stability of the dCas9 construct by Western Blot upon TMP addition?

- Fig 1B: here, and for all subsequent fluorescent images, quantification of fluorescence would be nice. For this particular figure, since the dCas9 construct is HA-tagged, the authors could show cell fractionation followed by Western Blot to show the difference in subcellular localization in a more clear/quantifiable way.

- Fig 1C: Fluorescence quantification would be nice

-Fig 1D: I am surprised that the mRNA fold increase is more or less the same for ORF50 (which is “induced” with the author CRISPRa system) and ORF57 and ORF45. I’m also surprised that at 72h post reactivation, there is so little ORF45 and ORF57 mRNA that should have been accumulating up to this point. How do the authors explain this? Here and in all bar graphs, errors bars need to be added as well as individual replicate points on the bars.

- Fig1C/1D: it would be nice to have a comparison with iSLK.219 cells that use the same KSHV genome – can this CRISPRa system be used in these cells? How does it compare to reactivation with Sodium Butyrate + Dox that the authors mention in the introduction?

- Fig 2B: place the NT bar closest to the Y axis for consistency with all the other bar graphs. Add errors bars and individual points like in the other figures.

- Fig 2C: how come the sg50-2 now increases ORF50 mRNA expression by 200 fold when it was only 20 fold in Fig 1D? Same question for ORF57: here sg50-2 triggers its expression (presumably through ORF50 expression) up to 100 fold when it was only ~15 fold in Fig1D.

- Fig 2E: I’m a bit confused: to quantify genome copy numbers, the authors extracted genomic DNA and ran a qPCR normalizing viral genomes over actin. However, it seems that the “viral primers” used by the authors are the same primers used to quantify K8.1 mRNA expression. Primers for viral genome copy numbers needs to target non-coding region of the viral genome, like promoter regions. The authors should clarify/fix.

- Fig 3: I think that panel A, B, C and D could all be combined into one panel. Statistical significance is showed on panel F – it would be nice to see some similar statistical tests on the other panels (in this figure and others) because some of the fold changes appears small (for example: ORF50 mRNA sg50-2 versus  sg50-2+3).

- Comparison of Fig3E and 2D is really inconsistent: in 2D, sg50-2 induces really high expression of ORF50 while in 3E, ORF50 expression is faint and even lower (it seems) than ORF57 – how do the authors explain this?

- Fig 4A: needs quantification

- Fig 4D: authors should clarify how they quantified individual cells

- How do the authors explain that using sg50-2 has a stronger effect on reactivation than ectopic expression of myc-ORF50 when in fact they detect more cells expression + higher levels of myc-ORF50? This is a big incentive to use their system over the ectopic expression of ORF50 so the authors should try to explain this difference or at least provide some perspectives on this in the discussion.

-“we could be able to identify novel sites where the KSHV lytic replication cycle is triggered spontaneously, 
possibly representing the viral reservoirs” (line 333-335): I think that the results presented here do not support this claim. What this article shows is that we could artificially trigger KSHV replication cycle using CRISPRa which is a great tool but this would in no way provide any more information of natural reservoirs. However, I think that the strength of this system could be that this can easily be used in already established cell lines or in new cell lines (user friendly) to replace current disruptive methods (sodium butyrate, TPA...) and/or to specifically trigger expression of one viral gene in the context of the full replication cycle. So the author should re-adjust what is emphasized in the discussion for clarity.

Minor phrasing problems:

line66: “is devoid of the nuclease activity” – remove “the” 

line 69: “allowed modulation of the targeted gene expression” 
-- remove “the”

line189: symbol problem

line 189: remove “and” before RFP and add it before “a viral lytic promoter”

line 290: “This indicates
 that despite of the higher levels of ORF50”  
-- remove “of the”

line 319: “When designing this study...” consider rewording this sentence for clarity

Author Response

Reviewer #3

In this study, the authors use a previously designed CRISPRa system to trigger KSHV reactivation. They show that this system works well to either target individual viral genes or trigger the whole reactivation cascade by inducing ORF50 expression. To my knowledge, it is the first time that someone tried to use CRISPRa on the KSHV genome. This is a potentially important new tool that could be of interest to the KSHV community.

Response: We thank the reviewer for finding our work novel and of general interest for the field. Below is our point by point response to the reviewer’s comments.

However, to better support their claims, the authors should address the following points:

Point 1: The authors use TMP to send their dCas9 construct to the nucleus where it will presumably act on the viral genome. It is unclear from the manuscript but I would assume that the cell line they further create (combining dCas9 expression and the KSHV genome) would have to be under constant TMP condition? If so, the authors should include a control where they use TMP alone (no sgRNA) and see the effect on viral gene expression.

Response to point 1: Actually, there is no need to grow these cells under constant exposure to TMP. TMP was added only during the experiment to increase DD-dCas9-VP192 stability which allows efficient accumulation to the nucleus. In addition, non-transfected (NT) control cells were also treated with TMP in the experiments shown here.

Point 2: Still on TMP: In the articles cited (Ref 21 and 22), TMP is used to increase stability of the construct but here the author show that it’s the localization that is impacted by TMP (Fig 1B) – how do the authors explain this difference? Do the authors see any difference in stability of the dCas9 construct by Western Blot upon TMP addition?

Response to point 2: We think that by increasing the half-life of the protein, TMP allows for better accumulation of DD-dCas9-VP192 to the nucleus. When unstable, DD-dCas9-VP192 could be degraded even before entering the nucleus, which could explain the low nuclear localization in DMSO treated cells in Fig 1B. We can also see by WB stabilization of the protein upon addition of TMP. A western blot analysis of lysates from cells treated with DMSO or TMP is now shown in the new fig 1B.

Point 3: Fig 1B: here, and for all subsequent fluorescent images, quantification of fluorescence would be nice. For this particular figure, since the dCas9 construct is HA-tagged, the authors could show cell fractionation followed by Western Blot to show the difference in subcellular localization in a more clear/quantifiable way.

Response to point 3: We now show quantification of HA (DD-dCas9-VP192) signal in Fig 1B. Particularly, we quantified DD-dCas9-VP192 intensity only within the area occupied by the nucleus. We think that performing cellular fractionation and immunoblot analysis is not necessary as IFA shows clearly a significant accumulation of the DD-dCas9-VP192 protein in the nuclei upon addition of TMP.

Point 4: Fig 1C: Fluorescence quantification would be nice

Response to point 4: Images in Fig 1C are random representative images from an automated high-content microscope. We think that quantification of these images is not needed since their main purpose is just to give the reader an impression of how cultures look in each treatment condition. In our opinion, quantification of viral lytic transcripts by RT-qPCR is a more accurate way to measure KSHV reactivation than indirect quantification of RFP signal.

Point 5: Fig 1D: I am surprised that the mRNA fold increase is more or less the same for ORF50 (which is “induced” with the author CRISPRa system) and ORF57 and ORF45. I’m also surprised that at 72h post reactivation, there is so little ORF45 and ORF57 mRNA that should have been accumulating up to this point. How do the authors explain this? Here and in all bar graphs, errors bars need to be added as well as individual replicate points on the bars.

Response to point 5: We thank the reviewer for pointing this out. A similar situation we have observed also in KSHV infected lymphatic endothelial cells where KSHV is reactivated spontaneously (PMID: 32518203). ORF50 and ORF45 were upregulated with similar fold difference compared to uninfected cells. Possibly, DD-dCas9-VP192 serves just to ignite the ORF50 gene expression and afterwards, ORF50 protein takes over the control of its own promoter and the rest of the lytic genes.

We have repeated this experiment including more biological replicates. All graphs now contain individual replicate points, error bars and SD.

Point 6: Fig1C/1D: it would be nice to have a comparison with iSLK.219 cells that use the same KSHV genome – can this CRISPRa system be used in these cells? How does it compare to reactivation with Sodium Butyrate + Dox that the authors mention in the introduction?

Response to point 6: We chose not to compare our cell line to iSLK.219 as we are not suggesting that this CRISPRa system should replace the use of iSLK.219 which is a great system to study the lytic cycle. Rather, we are suggesting that CRISPRa systems can further complement the host-pathogen interaction studies as now, control of both viral and cellular gene expressions is possible in a more physiological manner.

Point 7: Fig 2B: place the NT bar closest to the Y axis for consistency with all the other bar graphs. Add errors bars and individual points like in the other figures.

Response to point 7: We have repeated the experiment for Fig 2B with more biological replicates. Data are now shown in the new Fig 2B with error bars and individual data points. NT now is closest to the Y axis.

Point 8: Fig 2C: how come the sg50-2 now increases ORF50 mRNA expression by 200 fold when it was only 20 fold in Fig 1D? Same question for ORF57: here sg50-2 triggers its expression (presumably through ORF50 expression) up to 100 fold when it was only ~15 fold in Fig1D.

Response to point 8: These experiments were done at different times. In Fig 1D transfection efficiency apparently was not optimal. We have repeated this experiment again and new data indicating efficient ORF50 and ORF57 upregulation upon sg50-2 are now shown in the new Fig 1D. ORF50 is upregulated by 100-fold, ORF57 by 60 fold, ORF45 by 100 fold and K8.1 by about 400-fold which is similar to experiments shown in other figures.

Point 9: Fig 2E: I’m a bit confused: to quantify genome copy numbers, the authors extracted genomic DNA and ran a qPCR normalizing viral genomes over actin. However, it seems that the “viral primers” used by the authors are the same primers used to quantify K8.1 mRNA expression. Primers for viral genome copy numbers needs to target non-coding region of the viral genome, like promoter regions. The authors should clarify/fix.

Response to point 9: The K8.1 primers, used here, are also suitable for quantifying genome copies by qPCR. The PCR product is about 200 bp, so within the range for qPCR analysis (<500bp). Regarding actin, we are using primers that bind to the human DNA region where actin gene is found. In addition, during DNA isolation, RNA is degraded by RNaseA treatment therefore, there is no need to use primers that target a non-coding region.

- Fig 3: I think that panel A, B, C and D could all be combined into one panel. Statistical significance is showed on panel F – it would be nice to see some similar statistical tests on the other panels (in this figure and others) because some of the fold changes appears small (for example: ORF50 mRNA sg50-2 versus  sg50-2+3).

Response: We agree that some of the differences appear small in the old Fig 3 A-D panels. To predict significant differences more accurately, we have added three more repeats. One-way ANOVA followed by Dunnett’s multiple comparison test, shows a significant increase in all measured transcripts in sg50-2+3 groups as compared to sg50-2 and sg50-3. A-D panels were combined as suggested. We show now statistics also in Fig 1B and E and Fig 4 C-D.

- Comparison of Fig3E and 2D is really inconsistent: in 2D, sg50-2 induces really high expression of ORF50 while in 3E, ORF50 expression is faint and even lower (it seems) than ORF57 – how do the authors explain this?

Response: In the new Fig 3B (corresponding to the old 3E), the contrast was modified so that bands appear more appropriately. In fig 2D we chose deliberately a high ORF50 exposure to show that upon sg57-4 expression, so ORF57 expression, there was no ORF50 production.

- Fig 4A: needs quantification

Response: We find WB and virus titration as more accurate methods to quantitatively show the differences in virus reactivation. Moreover, in this particular case quantifying RFP would be misleading. Being expressed from the PAN, an ORF50 dependent lytic promoter, RFP is higher in cells overexpressing the Myc-ORF50. However, as found later, this is not an indication of a more efficient reactivation. Therefore, we prefer to leave the images without quantification to be used only as representatives for the reader to have an idea how the cultures looked in each condition.

- Fig 4D: authors should clarify how they quantified individual cells

Response: In the methods section 2.5, last sentence, we have added the functions that were used to identify nuclei and measure intensity in individual cells using the Cell Profiler 3.0 software package.

- How do the authors explain that using sg50-2 has a stronger effect on reactivation than ectopic expression of myc-ORF50 when in fact they detect more cells expression + higher levels of myc-ORF50? This is a big incentive to use their system over the ectopic expression of ORF50 so the authors should try to explain this difference or at least provide some perspectives on this in the discussion.

Response: We thank the reviewer for the question. Our data indicates that more ORF50 does not lead to a better reactivation when provided ectopically in the system. Our current hypothesis is that the dCas9-activator leads to a better opening of the virus chromatin at the ORF50 promoter that facilitates the expression of nearby viral genes which, in turn, leads to more efficient viral replication, packaging and egress. Only after these processes have been studied further in detail, we can have the first clues to explain why the higher ectopic expression of ORF50 leads to poorer reactivation than triggering the ORF50 expression from KSHV genomes with dCas9-activators. A perspective, is now provided in the Discussion section, paragraph 3.

-“we could be able to identify novel sites where the KSHV lytic replication cycle is triggered spontaneously, 
possibly representing the viral reservoirs” (line 333-335): I think that the results presented here do not support this claim. What this article shows is that we could artificially trigger KSHV replication cycle using CRISPRa which is a great tool but this would in no way provide any more information of natural reservoirs. However, I think that the strength of this system could be that this can easily be used in already established cell lines or in new cell lines (user friendly) to replace current disruptive methods (sodium butyrate, TPA...) and/or to specifically trigger expression of one viral gene in the context of the full replication cycle. So the author should re-adjust what is emphasized in the discussion for clarity.

Response: Thank you for the suggestion. We have removed this sentence in the discussion and added instead a paragraph focusing on the advantages of using CRISPRa system to trigger the KSHV lytic cycle (paragraph 3).

Minor phrasing problems:

Line66: “is devoid of the nuclease activity” – remove “the” 

Response: We removed “the” in line 66

line 69: “allowed modulation of the targeted gene expression” 
-- remove “the”

Response: “the” was removed from line 69

line189: symbol problem

Response: The α symbol was added, new line 201.

line 189: remove “and” before RFP and add it before “a viral lytic promoter”

Response: We modified the sentence in line 201 as “This virus expresses EGFP from a cellular constitutive promoter (EF1α) and RFP from the viral lytic PAN promoter that is under the control of ORF50”.

line 290: “This indicates
 that despite of the higher levels of ORF50”  
-- remove “of the”

Response: “of the” was removed, line 321 revised manuscript. The sentence in the revised manuscript is now “This indicates that despite higher levels of ORF50, these cells did not undergo a more efficient, productive KSHV lytic replication (Figure 4D)”.

line 319: “When designing this study...” consider rewording this sentence for clarity

Response: the sentence starting now in line 351 in the revised manuscript was changed to “Therefore, by directing the binding of the large DD-dCas9-VP192 protein (>250 kb) downstream of the RBP-Jk bindings sites could, in principle, block the RNA polymerase to transcribe the region of ORF50 downstream of the dCas9 binding site.”

Round 2

Reviewer 1 Report

The authors have addressed reviewer concerns with additional data, experimental repeats and text revisions. Overall, the manuscript is much improved.